# Relationship between Thought Style, Emotional Response, Post-Traumatic Growth (PTG), and Biomarkers in Cancer Patients

**DOI:** 10.3390/ijerph21060763

**Published:** 2024-06-13

**Authors:** Mariana Sierra-Murguía, Martha L. Guevara-Sanginés, Gabriela Navarro-Contreras, Guillermo Peralta-Castillo, Amalia Padilla-Rico, Lucía González-Alcocer, Ferrán Padrós-Blázquez

**Affiliations:** 1Cancer Center Tec100, Ignacio Zaragoza 263 H16, Col. Centro, Querétaro 76000, QE, Mexico; gperalta@cancercentertec100.com (G.P.-C.); apadilla@cancercentertec100.com (A.P.-R.); lgonzalez@cancercentertec100.com (L.G.-A.); 2Economic Administrative Science Division, University of Guanajuato, Fraccionamiento 1, Col. El Establo S/N, Guanajuato 36250, GJ, Mexico; leticiag@ugto.mx; 3Health Science Division, University of Guanajuato, Blvd. Puente Milenio #1001, Fracción del Predio San Carlos, León 37670, GJ, Mexico; g.navarro@ugto.mx; 4Psychology Faculty, Universidad Michoacana San Nicolás de Hidalgo, Francisco Villa 450, Col. Dr. Miguel Silva, Morelia 58120, MC, Mexico; fpadros@uoc.edu

**Keywords:** post-traumatic growth, lymphocyte-neutrophil ratio, monocyte- lymphocyte ratio, cancer, thought styles

## Abstract

(1) Introduction: Cancer diagnosis has been related to depression, anxiety, and distress, as well as to post-traumatic growth (PTG). One of the mediating variables for emotional response is thought style (rumination, cognitive avoidance, and cognitive engagement). (2) Aim: To identify the relationship between thought style and emotional responses to cancer. A secondary aim was to identify the relationship between emotional responses and inflammatory immunological biomarkers. (3) Method: A total of 115 patients with cancer were included in the study. Before initiating cancer treatment, patients were assessed using the Hospital Anxiety and Depression Scale (HADS), distress thermometer, and Post-Traumatic Growth Inventory (PTGI). Patients provided their most recent blood biometry. (4) Results: Rumination correlated with anxiety, depression, and distress. Cognitive avoidance correlated with PTG (−0.240) and distress (−0.209). Cognitive engagement correlated with PTG (0.393). Regarding thought style and biomarkers, a negative correlation was observed for absolute neutrophils with cognitive avoidance (−0.271) and rumination (0.305). Regarding biomarkers and emotional responses, there was a negative correlation between PTG and absolute lymphocytes (−0.291). There was also a correlation between PTG and neutrophils (0.357) and neutrophil-to-lymphocyte ratio (NLR) (0.295). (5) Conclusions: Thought style is related to the emotional response to a cancer diagnosis; rumination is related to depression, distress, and anxiety; and cognitive engagement is related to PTG. PTG is related to inflammation and immunological biomarkers.

## 1. Introduction

Cancer is a leading cause of mortality worldwide [1]. The experience of cancer represents an important stressor, since it involves a group of debilitating, chronic diseases that entail a lethal risk, which, in turn, is accompanied by a series of aversive events (diagnosis, treatment, and side effects) [2]. The diagnosis and treatment of cancer frequently lead to stress, emotional discomfort (distress), difficulties in adjusting to the disease, and a decrease in social interactions [3]. Cancer has a high psychological impact, and emotional discomfort (distress), depression, anxiety, and post-traumatic stress are the most studied and reported disorders in this population [4].

In addition to these emotional responses, positive changes can be triggered in cancer patients [5], meeting the criteria for post-traumatic growth (PTG), which, according to the Calhoun and Tedeschi model, is defined as positive psychological changes that may occur as a result of facing highly stressful life events [6]. There is an explanatory model for the development of PTG that begins when a person is exposed to a highly stressful event, such as the diagnosis of cancer, which is related to the experience of a high degree of emotional discomfort. When a person presents a thought style, such as cognitive engagement, which is associated with problem solving and searching for meaning, he/she is more likely to develop PTG [6].

These positive changes can be observed in various domains: self-concept (assessing one’s own strength and resilience), appreciation of new possibilities in life, social relations (increasing the sense of closeness with people, especially friends and family), life philosophy (change in values and life priorities), and spirituality (not limited to involvement in religious activities, but this could be an example).

In cancer patients, PTG has been associated with better adaptation to the disease, and thus with reports of better physical and mental health, fewer symptoms of distress and post-traumatic stress, and a greater number of healthy behaviors and better therapeutic adherence to oncology treatments [7,8]. Post-traumatic growth has also been associated with a change in health behaviors (which in turn are associated with a lower risk of relapse and greater survival), increasing healthy eating, and physical exercise in survivors. These changes are related to “changes in the philosophy of life”, as reported in the Post-Traumatic Growth Inventory [7].

Among the factors related to PTG in patients with cancer, sociodemographic characteristics, coping styles, thought styles, and social support have been reported [8].

Regarding coping styles, positive reframing (changing the perception towards something more positive) and problem-solving approach were positively associated with PTG. Similarly, these cognitive and behavioral responses (thought and coping styles) can produce different physiological responses and are influenced by thought style [9].

A process-oriented framework related to the emotional response to cancer has been noted as being a necessity for psycho-oncological treatment, because of the knowledge of the relationship between thought styles and emotional responses, as well as the disease’s trajectory, and the design, selection, and implementation of emotional regulation strategies [10].

Thought style has been defined as a mediating variable for emotional response to cancer. The process of thinking carefully, repeatedly, or frequently about oneself and one’s context is a central element in different models of adjustment to illness (adaptive or maladaptive). This has been found to occur in pathological contexts (for example, rumination in depression), but also in normative contexts [11], and it has been suggested that their specific content can predict adjustment to a stressor [12].

Cognitive avoidance can be found within thought styles, and refers to actions that are carried out with the aim of avoiding exposure to thoughts that generate emotional discomfort. Cognitive avoidance in cancer (for example, denial of the diagnosis or avoidance of thoughts related to cancer) does not allow the mental processing of a stressful experience to occur, and is associated with an increased perception of physical symptoms associated with treatment in patients with cancer and greater emotional distress in both the medium and long terms [13,14]. Cognitive avoidance, like behavioral avoidance, produces a temporary reduction in anxiety in the short term but prevents coping with fears and, consequently, their emotional processing [15]. Therefore, it is considered beneficial for some cancer patients during critical periods [14].

Rumination is another thought style, described as an emotional regulation strategy that consists of intrusive, repetitive thoughts that immediately generate emotional distress. Rumination is considered a maladaptive thought style because it is a predictor and exacerbator of psychopathology [16] and interferes with the ability to solve problems and improve moods [16].

One of the ways in which repetitive thoughts can be adaptive is through emotional processing and reflection. In the post-traumatic growth model [6], the term cognitive engagement is used, emphasizing that it is frequently thought that it is not necessarily intrusive and that it includes memories, the search for solutions to problems, and attempts to find meaning during a stressful situation [17]. Meaning-seeking thoughts have been described as an effective coping strategy to overcome stressful life experiences [18].

Thus, a priori cognitive engagement can reduce stress associated with cancer and generate changes in the immune response and survival of patients. In cancer patients, post-traumatic growth has been shown to be related to a greater balance at the physiological level, specifically with the blood leukocyte count and its return to basal levels after chemotherapy [19].

Dunigan [19] carried out a study that aimed to determine the relationship between post-traumatic growth, immune response, and survival in patients with hepatocarcinoma using the blood count of peripheral leukocytes (as a marker of immunity) and survival. They found a significant relationship between PTG and an elevated lymphocyte count, and the spirituality subscale was associated with a higher number of peripheral leukocytes in both sexes. In the same study, patients who presented with post-traumatic growth survived an average of 186 days longer than those in the group with the lowest PTG.

In cancer treatment, the outcome is no longer determined only by the characteristics of the tumor; the variables associated with the patient are also relevant. In recent years, inflammation associated with cancer has been found to be decisive for progression and survival in most patients; however, the systemic inflammatory response is associated with patient-related variables such as nutritional, functional, immune status, and psychological stress [20]. Likewise, low-grade systemic inflammation has been reported in some psychiatric disorders, including psychosis, mood disorders, and personality disorders [21]. Inflammation appears to play an important role in the development and progression of various types of cancers by promoting cell proliferation, angiogenesis, metastasis, and tumor response to systemic therapies [22]. Neutrophils, platelets, T cells, and B lymphocytes have been suggested to play important roles in tumor-associated inflammation and immunology [23]. Based on this, various inflammatory markers in the blood count (platelets, lymphocytes, and neutrophils), neutrophil/lymphocyte (N/L) ratio, platelet/lymphocyte (P/L) ratio, and mean platelet volume have been studied in malignant tumors. These indices have been described as highly replicable, inexpensive, and widely available markers of inflammation and immunology [24].

Schubert, Schmidt, and Rosner [25] conducted a systematic review of the relationship between post-traumatic stress and psychological and biological variables, and concluded that studies are needed regarding the impact of PTG on biological variables.

Regarding the relationship between thought style and biological variables, it has been identified that rumination is associated with greater inflammation, measured with C-reactive protein [26], and that this relationship is moderated by psychosocial factors [27]. Similarly, it has also been concluded that there is little evidence in this regard, so it has been suggested that more studies are needed on this relationship [27].

It could be suggested that thought style works as a mediating variable between the oncological diagnosis and the emotional response of the patient before the diagnosis and throughout the treatment. This relationship between thought style and emotional response has been previously studied, but only with a single thought style and rarely with cancer patients, for example, rumination for depression [28] or cognitive impairment for PTG [6], but it has not been addressed in the same evaluation (rumination, cognitive compromise, and avoidance in the same subject), nor have they been compared with each other, nor have they been studied in this way in cancer patients.

Therefore, the main objective of this work was to study the relationship between thought styles, emotional responses (post-traumatic growth, distress, anxiety, depression), and inflammatory immunological biomarkers (leukocytes, lymphocytes, neutrophils, monocytes, and platelets), and their corresponding ratios: INL, ILM, and IPL). Similarly, the relationship between emotional states and biomarkers was studied.

## 2. Materials and Methods

This was a descriptive, explanatory, and correlational study. Convenience sampling was used in this study. 

We expected to find moderate negative relationships between ruminative cognitive style and distress, anxiety, and depression. On the other hand, a negative relationship between avoidant style and distress was hypothesized, but we did not expect to find relationships between avoidant style and anxiety, depression, and post-traumatic growth. Finally, we expected to find a significant and positive relationship between cognitive impairment and post-traumatic growth.

### 2.1. Participants

The inclusion criteria were adult cancer patients who were about to start cancer treatment (radiotherapy and/or chemotherapy), agreed to participate in the study, and knew how to read and write. A total of 23 patients were excluded because they experienced a recurrence of the disease or had previously received cancer treatment (due to the influence that this could have on post-traumatic growth). Three patients who either did not complete the psychological evaluation questionnaires or did not sign the informed consent form were eliminated.

A total of 115 patients diagnosed with oncology, who were about to start their treatment in a private institution in the city of Querétaro, Mexico, were included.

The sex breakdown of the participants was 45 (39.1%) males and 70 (60.9%) females. The overall mean age was 52.8 years (SD = 14.50) and the participants’ ages ranged from to 18 to 83 years. The marital status of the participants was 20 (17%) single, 52 (45.2%) married, 4 (3.5%) widowed, 4 (3.5%) in a free union, and 1 (0.9%) divorced, and 34 (29.6%) did not report it.

The included patients had different types of cancer, as described in Table 1. The approximate time from diagnosis to the date of initial evaluation was, on average, 12.09 (SD = 10.53). The data related to the clinical stage are described in Table 2, and in terms of the type of treatment, in Table 3.

### 2.2. Instruments

#### 2.2.1. Distress Thermometer

In their guidelines for driving distress, the National Cancer Comprehensive Network (NCCN) [2] proposed the thermometer of distress as an instrument for screening for identifying emotional discomfort in patients with cancer. This instrument has been adapted and validated for the Mexican oncological population [29] and is trustworthy both for clinical use and research on the emotional impact of cancer. This instrument has a sensitivity of 93%, specificity of 76%, positive predictive value of 82%, and negative predictive value of 90%. The internal consistency was shown by Cronbach’s alpha = 0.87.

#### 2.2.2. Hospital Anxiety and Depression Scale (HADS)

Originally developed by Zigmond and Snaith [30], this scale is used to screen for depression and anxiety. It consists of 12 multiple-choice items, with global consistency shown by a Cronbach‘s alpha of 0.86, and 0.79 and 0.80 for each subscale (anxiety and depression). It was also validated in a Mexican oncological population [31].

#### 2.2.3. Tedeschi and Calhoun Post-Traumatic Growth Inventory 

This is a self-applied scale of 21 Likert-type items that measures five areas of growth after a stressful event. It had an alpha of 0.90, and the Spanish version for the Mexican population used by Guzmán Sescosse et al. was applied [32].

#### 2.2.4. Cancer-Associated Thought Styles Scale (IEPRAC)

This instrument was designed for this study. The final version has three subscales of five items each: rumination, cognitive engagement, and avoidance. The reported Cronbach’s alpha values were 0.87, 0.82, and 0.72, respectively [33].

### 2.3. Procedure

Upon identifying a patient who was going to start treatment (chemotherapy or radiotherapy), the medical oncologist referred the patient to the psycho-oncology department.

The patient was approached and the objectives and procedures of the study were explained. The approximate application time of the inventories was between 20 and 30 min per participant.

Before starting the corresponding medical oncology treatment, psychological variables (anxiety, depression, emotional discomfort, PTG, and social support) were evaluated.

The patient was asked for a copy of the blood count analysis closest to the date of psychological evaluation (regularly, these analyses were performed periodically during medical treatments to monitor the patient’s health status).

### 2.4. Data Analysis

For sociodemographic variables, initial results, and clinical status of patients (diagnosis time, diagnosis, stage, and treatment), descriptive statistics were used (mean and standard deviation for quantitative variables and frequencies and percentages for nominal and ordinal variables).

To determine the relationship between the thought style variables and PTG, anxiety, depression, and distress, a multiple linear regression analysis and a logit analysis were performed.

For the relationship between PTG and biomarkers, a Pearson correlation analysis was performed between the scores obtained in the inventory and the biomarkers (neutrophils, monocytes, lymphocytes, leukocytes, platelet count, neutrophil/lymphocyte ratio, and lymphocyte/platelets), as well as between depression and anxiety scores with the same biomarkers.

### 2.5. Ethical Considerations

The patients were informed verbally and through informed consent that they could voluntarily participate in the study, and about its nature and their right to not participate without affecting their cancer treatment. After this, patients who agreed to participate provided virtual informed consent. This was an observational study, so the risks were minimal; however, study approval was obtained from the ethics and research committee of the cancer center, registered with the number 22CI22014037.

## 3. Results

Regarding the relationship between thought styles and anxiety, depression, distress, and PTG, a slight positive correlation between cognitive engagement and PTG stands out. On the other hand, regarding rumination, significant positive correlations were obtained with distress (low), and anxiety and depression (moderate). Finally, a slight negative correlation was observed between avoidance and distress (Table 4).

### Thought Style, Emotional Response, and Biomarkers

Regarding the relationship between thought style and biomarkers, only a slight and significant negative correlation was identified between absolute neutrophil counts and avoidant and ruminative cognitive styles. No significant correlations were found between biomarkers and cognitive engagement (Table 5).

Regarding the relationships between emotional responses (anxiety, depression, distress, and PTG) and biomarkers, a slight negative correlation was identified between PTG and the percentage of lymphocytes and the absolute number of lymphocytes. The correlation between PTG and the percentage of neutrophils and the lymphocyte neutrophil index (NLI) was also significant. For the remaining emotional response variables and biomarkers, no significant correlations were found (see Table 6).

## 4. Discussion

The objective of the present study was to examine the relationship between thought style and emotional response (PTG, distress, anxiety, and depression) and inflammatory immunological biomarkers (leukocytes, lymphocytes, neutrophils, monocytes, and platelets, and their corresponding ratios INL, ILM, and IPL).

The rumination thought style showed a positive and moderate relationship with anxiety and depression and a low relationship with distress. This relationship has been reported previously in the literature [11,34]. Regarding the avoidant thought style, a slight and significant negative correlation was found with distress, and no correlation was found with the presence of symptoms of anxiety and depression. This result could be explained by the role of avoidance as a coping strategy, which is useful in the short term. Previous studies [14,15] have described that, initially, upon receiving a cancer diagnosis, it may be effective to use an avoidant thought style to a certain extent to deal with the disease and cancer treatment, especially in the early stages. To study the possible variability of the impact of the avoidant thought style on the levels of distress, anxiety, and depression over time, it is necessary to study the evolution of the mood of patients who showed high values of cognitive avoidance over time from treatment to a follow-up of several months.

It is possible that very high levels of avoidance style over time could have harmful effects, and both anxiety and depressive symptoms could increase. In fact, the negative and low correlation found between avoidance and PTG is relevant, and can be explained by the theory. For PTG to occur, it is necessary that the stressor event be interpreted as threatening, and therefore generates a certain degree of anxiety and emotional discomfort, to be able to activate the pertinent cognitive resources to face and reinterpret the stressor [35]. Therefore, what is adaptive and healthy is that the patient does not abuse cognitive avoidance to a certain extent and can experience a certain degree of discomfort when directly facing negative thoughts and emotions derived from the situation.

In relation to the third thought style, cognitive engagement, no relationship was observed with scores for distress, anxiety, and depression, but a positive and almost moderate correlation was observed with post-traumatic growth, which is in accordance with the theory of Calhoun and Tedeschi [6] on PTG, where they describe cognitive engagement as an element that favors PTG.

It is important to point out that in order to develop a cognitive compromise, which is related to PTG, a certain degree of emotional discomfort is necessary (at least in the initial phase) and that the person can learn to tolerate that emotional discomfort and direct the thought style towards it, search for meaning, and solve problems.

In contrast, a low correlation was observed between PTG and absolute lymphocytes, which are parameters for adaptive immunity and inflammation, and a positive correlation was observed with neutrophils. Based on these relationships between PTG, lymphocytes, and neutrophils, it can be inferred that PTG is related to an initial inflammatory response associated with the emotional discomfort that is generated in the first instance when facing a potentially traumatic situation. This is consistent with the theory of Calhoun and Tedeschi [6], who refer to the PTG process that begins with the perception of threat and emotional discomfort.

INL was negatively correlated with PTG, which is associated with an inflammatory response and chronic stress [36]. In cancer patients, it has been described that having an elevated INL (greater than 5) is associated with a worse prognosis of the disease, a larger tumor size, and having more advanced disease. In addition to this, it was also described that patients who manage to reduce INL at the end of treatment have better prognostic values than those who do not manage to reduce it [20]. This negative correlation between PTG and INL was not expected, nor were the relationships shown between neutrophils, lymphocytes, and PTG, since, by generating a “positive” cognitive change, a decrease in biomarkers associated with the presence of stress would be expected. However, it is likely that, according to the theory of Tedeschi and Calhoun [6], in order for PTG to occur, it is necessary to present some type of emotional discomfort in the first phase. This is also consistent with another finding obtained in the present study, which showed a positive correlation between PTG and distress. It would be pertinent to study the relationship between thought style and emotional responses longitudinally (pre- and post-treatment and follow-up). It is also suggested that those cases that present with PTG should be identified and to evaluate the inflammatory and immunological responses over a long period of time.

### Limitations

It should be noted that there was no control for variables that may affect the biomarkers used (diet, body mass index, smoking, exercise, comorbidities, and medications).

Another limitation is the heterogeneous sample used, in the sense that different types of cancer, times of illness, and clinical stages were included, and these variables can also affect both biomarkers and thought styles.

Likewise, the period in which the evaluations were carried out was during the COVID-19 pandemic; therefore, the emotional state could have been affected by this condition and not only by coping with cancer.

It is important to point out that results are correlations, which does not mean causality. This is a first approach to study this relation, but future research needs to be conducted in order to explain it. 

Future research should assess the change in the inflammatory–immunological ratios pre- and post-treatment, and relate it to both the thought style and emotional state of the patient. Similarly, greater control of confounding variables (diet, body mass index, medications, comorbid diseases, etc.) is suggested.

## 5. Conclusions

Thought styles correlate with emotional response to cancer. Rumination thought style showed a positive and moderate relationship with anxiety and depression. Regarding the avoidant thought style, a slight and significant negative correlation was found with distress and low symptoms of anxiety and depression. Cognitive engagement correlated with post-traumatic growth.

From the results obtained and the conclusions, it could be valuable to generate psychological interventions aimed at modifying the thought style, promoting cognitive commitment and post-traumatic growth, reducing rumination and the tendency to present depression and anxiety, and moderating the avoidant thought style.

## Figures and Tables

**Table 1 ijerph-21-00763-t001:** Type of cancer.

Diagnosis	Overall *n* (%) *n* = 115	Male *n* = 45	Female *n* = 70
Breast	38 (34.5)	1 (2.3)	37 (56.1)
Gastric	15 (13.6)	9 (20.5)	6 (9.1)
Urological	11 (9.5)	11 (24.4)	-
Hematological	10 (9.1)	6 (13.6)	4 (6.1)
Gynecological	8 (7.3)	-	8 (12.1)
Head and neck	6 (5.5)	4 (9.1)	
Lung	4 (3.6)	2 (4.5)	
Skin	4 (3.6)	2 (4.5)	
Other	14 (12.7)	9 (20.5)	5 (7.6)
Don’t know/Didn’t answer	5 (4.3)	1 (2.3)	4 (6.1)

**Table 2 ijerph-21-00763-t002:** Clinical stage.

		Clinical Stage				Unknown/NA
		**1**	**2**	**3**	**4**	
Sex	Male *n* = 45	1 (6.3)	3 (18.8)	1 (6.3)	11 (68.8)	29 (64.4)
	Female *n* = 70	7 (24.1)	4 (13.8)	9 (31)	9 (31)	41(58.7)
Total		8 (17.8)	7 (15.6)	10 (22.2)	20 (44.4)	70 (60.86)

*n* = 115.

**Table 3 ijerph-21-00763-t003:** Type of treatment.

Treatment	*n* (%)	Male *n* (%) *n* = 45	Female *n* (%) *n* = 70
Chemotherapy	32 (31.4)	15 (39.5)	17 (26.6)
Chemotherapy + Radiotherapy	7 (6.9)	2 (5.3)	5 (7.8)
Chemotherapy + Radiotherapy + Surgery	15 (14.7)	5 (13.2)	10 (15.6)
Chemotherapy + Radiotherapy + Surgery + Hormonal therapy	4 (3.9)	1 (2.6)	3 (4.7)
Surgery + Chemotherapy	20 (19.6)	7 (18.4)	13 (20.3)
Surgery + Radiotherapy	13 (12.7)	8 (12.5)	5 (13.2)
Other	11 (10.8)	3 (7.9)	8 (12.5)
Don’t know/Didn’t answer	13 (11.3)	4 (8.8)	9 (12.8)

*n* = 115.

**Table 4 ijerph-21-00763-t004:** Correlations between thought style and emotional response (distress, anxiety, depression, PTG).

	Distress	Anxiety	Depression	PTG
Cognitive Engagement	0.172	0.103	0.124	0.393 **
Rumination	0.346 **	0.567 **	0.545 **	0.016
Avoidance	−0.209 **	−0.033	0.006	−0.240 **

** *p* < 0.05 bilateral level. PTG: Post-traumatic growth

**Table 5 ijerph-21-00763-t005:** Correlations between thought style and biomarkers.

	Cognitive Engagement	Avoidance	Rumination
Leukocytes mm^3^	0.055	−0.056	−0.190
Lymphocytes %	−0.037	0.102	0.208
Absolute lymphocytes	−0.015	−0.034	−0.039
Neutrophils %	0.228	−0.175	−0.245
Absolute neutrophils	0.172	−0.271 *	−0.305 *
Monocytes %	−0.111	0.063	−0.036
Monocytes uL	−0.111	0.017	−0.023
Mean platelet volume fL	−0.119	0.023	−0.008
Platelet count	−0.020	0.024	−0.148
LMI	0.128	0.048	0.011
NLI	0.106	−0.060	−0.158
PLI	0.013	0.076	−0.079

* *p* < 0.01; N = 66. uL: microliter; fL: femtoliter; LMI: Leukocyte Monocyte Index; NLI: Neutrophile Lymphocyte Index; PLI: Platelet Lymphocyte Index.

**Table 6 ijerph-21-00763-t006:** Correlations between biomarkers and emotional response (anxiety, depression, distress and PTG).

	Distress	Anxiety	Depression	PTG
Leukocytes mm^3^	0.023	−0.112	−0.010	0.074
Lymphocytes %	0.168	0.161	−0.125	−0.280 *
Absolute lymphocytes	0.241	0.036	−0.076	−0.291 *
Neutrophils %	−0.134	−0.210	0.099	0.357 **
Absolute neutrophils	−0.009	−0.129	0.085	0.145
Monocytes %	0.013	0.004	0.080	0.041
Monocytes uL	0.194	−0.016	0.077	−0.034
Mean platelet volume fL	−0.076	0.116	0.147	−0.060
platelet count	−0.087	−0.103	−0.024	−0.119
ILM	−0.002	0.007	−0.065	−0.134
INL	0.004	−0.006	0.157	0.295 *
IPL	−0.059	−0.024	0.066	0.094

** *p* < 0.05. * *p* < 0.01.

## Data Availability

Data are unavailable due to privacy restrictions. If needed, contact the principal author (M.S.-M.). Data cannot be shared openly to protect participant privacy, but can be accessed by reaching the principal author (M.S.-M.) at msierra@cancercentertec100.com.

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
