# Peer review of "Relationship between Thought Style, Emotional Response, Post-Traumatic Growth (PTG), and Biomarkers in Cancer Patients"

_ijerph, 2024, doi:10.3390/ijerph21060763_

Round 1

Reviewer 1 Report

Comments and Suggestions for Authors

Thank you for the opportunity to review this manuscript. The manuscript “Relationship between thought style, emotional response, post-traumatic growth (PTG) and biomarkers in cancer patients” presents an important descriptive study. Considering the broader context of presented studies and further impact of the published papers, there are some corrections/suggestions that I have pointed out; suggested to be addressed.

1.      Abstract, Line 20-24: I would suggest adding analysis values of the results stated i.e., correlation values so that the readers understand how strong were the correlations. It is difficult to understand the impact of results without values.

2.      The authors have applied parametric statistical tests for analysis. Did the authors confirm the data normality for parametric tests?

3.      The authors state that that they have only obtained verbal inform consent. It would be good to mention for what reasons they could not obtain written inform consent.

4.      Considering different types of tumors, it would be good to provide some results of specific cancer types i.e., it is more stressful to have breast cancer than skin melanoma.

5.      Also (if possible), sensitivity analysis, based on different types of treatment.

6.      In the discussion section, I would suggest including results from studies conducted in different races and ethnic populations and compare results of the current study with them. According to literature, in many setting it is still a taboo to talk about cancer status with cancer patients. In some settings (with low educational status), cancer patients are not even informed about their disease and its status.   

Minor comments:

1.      I would suggest replacing “Cancer”, “Post-traumatic growth” and “thought styles” in key words with suitable similar words. As these words have already been used in the title of the manuscript.

Author Response

Correlation values were added in abstract.

Informed consent was applied verbally, written and explained to the patients, who also agreed to participate in study and signed to be included. This was included in ethical aspects in the paper. 

It has been described that emotional response is not mediated by type of cancer, this is why we did not include this analysis. 

Regarding studies in other races or ethnic populations, we have not found yet. 

Reviewer 2 Report

Comments and Suggestions for Authors

Overall I like this paper and the general thrust of it.

The study of Post Traumatic Growth is important.

But

The introduction and initial paragraphs of this paper are confusing and difficult to follow. Main reason for this is in relation to some confusion about abbreviations.

In line 45, the authors talk about development of PTC. There's no definition of PTC in the paper. This needs to be included.

On line 49 the authors refer to the development of CPT. Again, there's no definition. In trauma psychiatry CPT means Cognitive Processing Therapy. Please define these abbreviations.

The other issue I have with this paper is in the in the discussion the authors need to have a paragraph clearly entitled limitations. The discussion is a bit confusing to read. So, correlation they need to stipulate doesn't mean causality in any way and they need to state that in the limitations of this study. So in line, 322 the authors talk a little bit about limitations. The authors should expand on the issue about correlation and causation. I do not think that the paper should be published in the way it has been written. I do think the limitations are very important to point out in this paper the findings some of the findings are useful. But I think the paper needs to be reworked and made much clearer.

Another limitation with this study is the actual stage of treatment the patient is in. So, for example if you find negative ruminations being associated with various biomarkers, it's quite important to know where they the patient is in their treatment pathway. For example they may have grieved for their loss of health and possibly their life or they may be starting their grieving process.

Psychological processes may determine the level of rumination and acceptance of their illness. Difference circumstances governing acceptance eg – resilience, personality etc, may need to be discussed in the paper. So, in conclusion, the paper needs to be clarified. I wouldn't reject it, but I would ask for these clarifications to be completed.

Comments on the Quality of English Language

The English is generally good - there are a few repetitions which can be edited out I feel. 

Author Response

We are sorry, CPT is Spanish abbreviation of PTG. We have corrected it in the manuscript.

We have also added a limitations section including that it is only correlation, not meaning causality. 

Other factors, such as personality and resilience could be mediators of emotional response to cancer, but it is not the aim of the study, and have been mentioned in limitations because they were not controlled. 
